# Corneal confocal microscopy demonstrates minimal evidence of distal neuropathy in children with celiac disease

Hoda Gad[1], Saras Saraswathi[2○], Bara Al-Jarrah[2○], Ioannis N. Petropoulos[1○], Georgios Ponirakis[1○], Adnan Khan[1○], Parul Singh[3○], Souhaila Al Khodor[3‡], Mamoun Elawad[2‡], Wesam Almasri[2‡], Hatim Abdelrahman[2‡], Khalid Hussain[4‡], Mohamed A. Hendaus[5‡], Fatma Al-Mudahka[2‡], Khaled Abouhazima[2‡], Paraic McGrogan[2‡], Rayaz A. Malik[1,6‡]*, Anthony K. Akobeng[2‡]

1 Department Medicine, Weill Cornell Medicine-Qatar, Doha, Qatar, 2 Division of Gastroenterology, Hepatology, and Nutrition, Sidra Medicine, Doha, Qatar, 3 Research Department, Sidra Medicine, Doha, Qatar, 4 Division of Endocrinology, Sidra Medicine, Doha, Qatar, 5 General Pediatrics Department, Sidra Medicine, Doha, Qatar, 6 Institute of Cardiovascular Medicine, University of Manchester, Manchester, United Kingdom

○ These authors contributed equally to this work.
‡ These authors also contributed equally to this work. And RAM and AKA are joint senior authors and PI's on this work
* ram2045@qatar-med.cornell.edu

**Data Availability Statement:** All relevant data are within the manuscript and its Supporting Information files.

## Abstract

### Objectives

The aim of this study was to utilise corneal confocal microscopy to quantify corneal nerve morphology and establish the presence of sub-clinical small fibre damage and peripheral neuropathy in children with celiac disease.

### Methods

This is a cross-sectional cohort study of twenty children with celiac disease and 20 healthy controls who underwent clinical and laboratory assessments and corneal confocal microscopy. Corneal nerve fiber density (no.mm$^2$), corneal nerve branch density (no.mm$^2$), corneal nerve fiber length (mm.mm$^2$), corneal nerve fiber tortuosity and inferior whorl length (mm.mm$^2$) were quantified manually.

### Results

Corneal nerve fiber density (34.7±8.6 vs. 32.9±8.6; $P = 0.5$), corneal nerve branch density (47.2±24.5 vs. 47.3±20.0; $P = 0.1$) and corneal nerve fiber length (20.0±5.1 vs. 19.5±4.5; $P = 0.8$) did not differ between children with celiac disease and healthy controls. Corneal nerve fiber tortuosity (11.4±1.9 vs 13.5±3.0; $P = 0.01$) was significantly lower and inferior whorl length (20.0±5.5 vs 23.0±3.8; $P = 0.06$) showed a non-significant reduction in children with celiac disease compared to healthy controls. Inferior whorl length correlated significantly with corneal nerve fiber density ($P = 0.005$), corneal nerve branch density ($P = 0.04$), and corneal nerve fiber length ($P = 0.002$).

**Funding:** This publication was made possible by a Sidra Internal Research Fund (SIRF) and a Biomedical Research Program [BMRP-5726113101] grant from the Qatar National Research Fund (QNRF). The statements made herein are solely the responsibility of the authors.

**Competing interests:** The authors have declared that no competing interests exist.

## Conclusion

Corneal confocal microscopy demonstrates minimal evidence of neuropathy in children with celiac disease.

## Introduction

Celiac disease (CD) occurs in approximately 0.2–5.5% of children [1]. CD is classically associated with gastrointestinal (GI) symptoms such as diarrhea, abdominal pain, malabsorption and weight loss. However, there are an increasing proportion of patients with CD who present with extraintestinal manifestations involving the musculoskeletal, cardiovascular and neurological systems and present with short stature, anemia, osteoporosis and arthritis [2].

Neurological manifestations including cerebellar ataxia, gluten encephalopathy, sensorineural hearing loss, headache and cognitive decline occur in about 10% of adults with CD [3]. A recent systematic review established that up to 39% of adults with CD may have gluten neuropathy [4]. Luostarinen et al demonstrated a chronic axonal neuropathy using quantitative needle electromyography (EMG) in 23% of patients with CD [5]. Peripheral neuropathy may develop independent of enteropathic symptoms or lack of adherence to a gluten free diet (GFD) [6]. Chin et al, found a small fiber neuropathy characterized by burning, tingling and numbness in the hands and feet and diffuse paraesthesiae involving the face, trunk, and lumbosacral region in 40% of adults with CD, mildly abnormal electrophysiology in 90% and a mild-to-severe axonopathy in nerve biopsies [7]. In a study of 13 CD patients, 8 had neuropathic pain, 6 had cerebellar ataxia, 1 had white matter abnormalities on magnetic resonance imaging (MRI) of the brain and 1 had ataxia and cortical myoclonus [8]. A nationwide population-based study of patients with biopsy verified-CD reported an absolute risk of neuropathy of 64 per 100,000 person years in CD patients compared to 15 per 100,000 in matched controls; with an estimated risk of neuropathy being highest in the early years of diagnosis with CD (HR 4.4; 95% CI 2.6–7.4; $P < 0.001$) [9].

There are limited data showing evidence of peripheral neuropathy in children with CD. In a study of 27 children with CD, 3 (11%) had sub-clinical neuropathy with axonal neuropathy on electromyography in 2 (7%) and prolonged somatosensory evoked potentials in 1 patient [10]. Neurophysiological studies are highly sensitive, but are not easily performed in children [11]. Quantitative sensory testing to evaluate vibration and thermal perception thresholds are easy to perform, but lack reproducibility, especially in children and skin or nerve biopsies are invasive procedures [12]. Corneal confocal microscopy (CCM) is a rapid, non-invasive and well-tolerated ophthalmic imaging technique that has been used to objectively quantify neurodegeneration in adults with metabolic [13–20], cerebrovascular [21–24] and central neurodegenerative [25–28] diseases. We have also shown a significant reduction in corneal nerve fibre density, branch density and length in children and adolescents with type 1 diabetes mellitus (T1DM) [29,30] and progression over time [31]. CCM is a well-tolerated technique for the early detection of neuropathy in children with T1DM [32]. The aim of this study was to utilise CCM to quantify corneal nerve morphology in the central cornea and inferior whorl to establish the presence of sub-clinical small fibre damage and peripheral neuropathy in children with CD.

## Methods

Twenty participants with CD and 20 age-matched healthy controls underwent CCM in the outpatient department of Sidra Medicine between January 2019 to October 2019. Patients with a history of any other cause of neuropathy, malignancy, deficiency of B12 or folate, chronic renal, liver failure, connective tissue or systemic disease, previous corneal trauma or systemic disease affecting the cornea, ocular surgery, LASIK and a history of or current contact lens wear were excluded from the study. Both eyes were examined, and it took around 10 minutes to capture central and peripheral corneal nerve images. Clinical and metabolic data were collected from the nearest clinic visit and healthy participants did not undergo any clinical tests. All participants provided written assent and parental informed consent and the research adhered to the tenets of Declaration of Helsinki and was approved by Sidra Medicine (IRB 1500758–3) and Weill Cornell Medicine (IRB 17–00032) Research Ethics Committee.

### Corneal confocal microscopy procedure

Using the Heidelberg Retina Tomograph Cornea Module (Heidelberg Engineering, Germany), all participants underwent CCM. Two eyes were anaesthetized with 2 drops of Bausch & Lomb Minims® (Oybuprocaine hydrochloride 0.4% w/v). Oybuprocaine hydrochloride drops are used to numb the eyes to limit irritation or discomfort during the procedure. A drop of hypotears gel (Carbomer 0.2% eye gel) was placed on the tip of the objective lens and a sterile disposable TomoCap was placed over the lens, allowing optical coupling of the objective lens to the cornea to focus on the sub basal nerve plexus (SBNP). Images were taken by a single investigator (HG) to capture several layers of the cornea; epithelium, SBNP, stroma, endothelium, and inferior whorl.

### Image selection and quantification

Six images overlapping by no more than 20% were selected from the central cornea based on the quality of images, absence of artifacts such as pressure lines and other layers as well as out of focus images. Corneal nerve fiber density (CNFD) (fibres/mm$^2$) corneal nerve branch density (CNBD) (branches/mm$^2$) and corneal nerve fiber length (CNFL) (mm/mm$^2$) were quantified manually and corneal nerve fiber tortuosity (CNFT) was quantified automatically from the trace of the main corneal nerve fibres using CCMetrics (Fig 1A and 1B). An additional six images centred on the inferior whorl and immediately adjacent areas not overlapping by no more than 20% were selected and the inferior whorl length (IWL) (mm/mm$^2$) was quantified utilizing the manual CNFL mode in CCMetrics [33] (Fig 1C and 1D). The investigator was blind to the study group when performing CCM and analysing CCM images.

### Sample size and statistical analysis

The primary outcome measure was corneal nerve fiber pathology as assessed with CCM. A difference in nerve fiber length of 2.7 mm/mm$^2$ has been demonstrated to be a clinically significant difference representing the difference between mean nerve fiber length in controls and individuals with mild diabetic neuropathy [34]. The standard deviation for nerve fiber length (mm/mm$^2$) in controls has been reported to be 0.88 [35]. Using these parameters, a power of 0.80 and a two-tailed alpha of 0.05, a minimum sample size of 16 was calculated per group. Given the lack of precision associated with this calculation we increased the sample size to 20 per group to improve the power of the study.

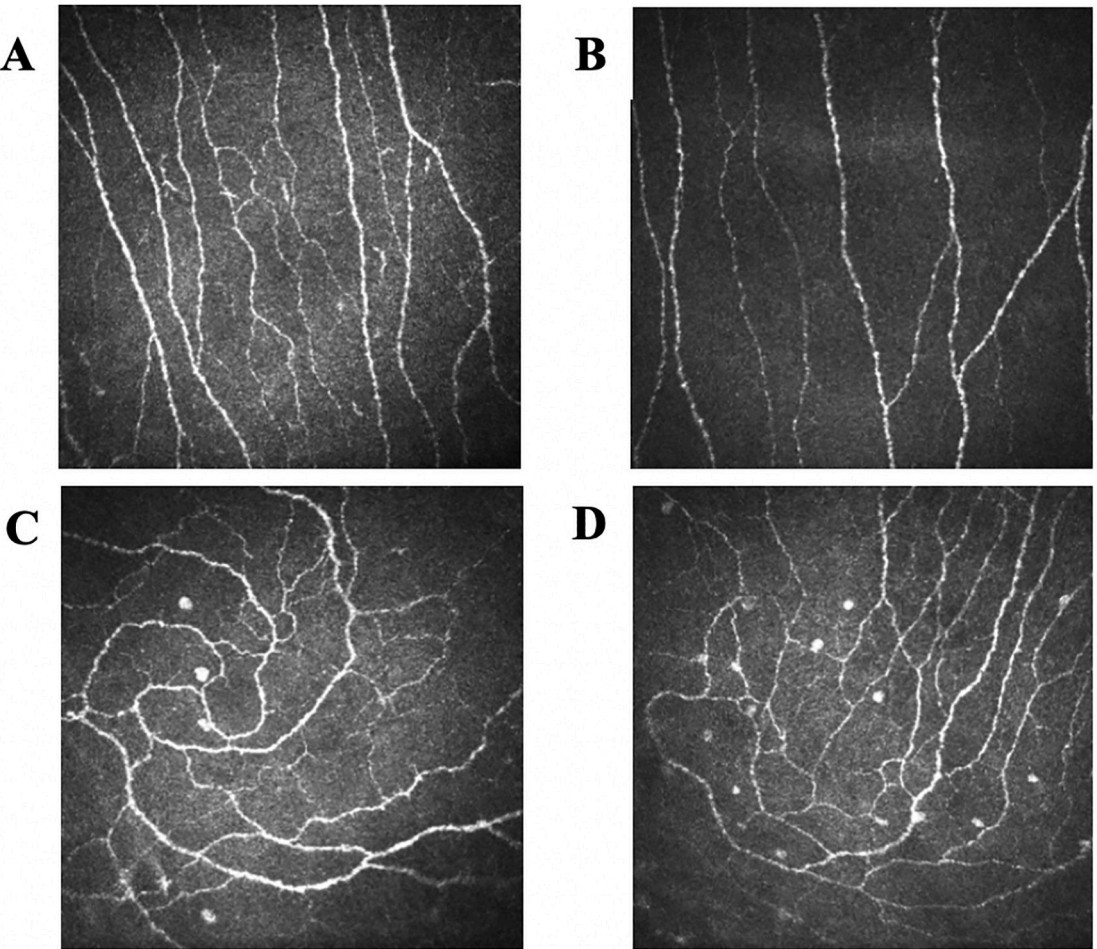

**Fig 1. CCM images of the central cornea in a healthy control (A) and child with celiac disease (CD) (B), and CCM images of the inferior whorl in a healthy control (C) and child with CD (D).**

All statistical analyses were performed using IBM SPSS Statistics software Version 26 and P<0.05 was considered statistically significant. Normally distributed data were expressed as mean ± standard deviation and the means were compared using an independent sample t-test. To investigate the association between clinical parameters and corneal nerve fibres parameters, Pearson correlation was used. Graph prism version 8 was used to build dot plots.

## Results

Clinical demographics are summarized in Table 1. The body mass index (BMI) (18.82±3.90 vs. 22.26±5.47; $P$ = 0.031) and 25 OHD (25-hydroxyvitamin D) (43.50±13.36 vs. 59.77±22.45; $P$ = 0.014) levels were significantly lower in children with CD compared to controls. There was no difference in CNFD (34.7±8.6 vs. 32.9±8.6; $P$ = 0.5), CNBD (47.2±24.5 vs. 47.3±20.0; $P$ = 0.1) and CNFL (20.0±5.1 vs. 19.5±4.5; $P$ = 0.8) (Fig 2A–2C). CNFT (11.4±1.9 vs 13.5±3.0; $P$ = 0.01) was significantly lower in children with CD compared to controls (Fig 2D). IWL demonstrated a non-significant reduction in patients with CD compared to healthy controls (20.0±5.5 vs 23.0±3.8; $P$ = 0.1) (Fig 2E). IWL was >2SD lower than the mean of controls in 20% of children with CD.

**Table 1. Clinical demographics and laboratory measures in control subjects and CD patients.**

|  | Healthy (n = 20) Mean ± SD | CD (n = 20) Mean ± SD | *P*-value |
| --- | --- | --- | --- |
| Age | 12.83±1.91 | 11.78±1.74 | 0.077 |
| Duration of disease | - | 4.49±4.02 | N/A |
| Height (m) | 1.44±0.13 | 1.38±0.14 | 0.125 |
| BMI (kg/m$^2$) | 22.26±5.47 | 18.81±3.90 | **0.031** |
| Tissue Transglutaminase antibody (U/mL) | 0.51±0.007 | 29.03±49.72 | 0.446 |
| Hemoglobin (g/dl) | 125.50±7.84 | 125.63±7.45 | 0.963 |
| Platelets x 10$^{\wedge 9}$/L | 333.75±73.20 | 325.42±77.51 | 0.768 |
| WBC x 10$^{\wedge 9}$/L | 7.72±1.98 | 6.73±1.89 | 0.171 |
| 25 OHD (ng/mL) | 59.77±22.45 | 43.50±13.36 | **0.014** |
| Vitamin B12 (ng/mL) | 180–914 | 321.84±89.40 | N/A |
| Folic acid (nmol/L) | 11.3–47.6 | 24.977±14.30 | N/A |
| Serum Iron (μmol/L) | 13.65±2.76 | 11.46±7.67 | 0.700 |

Data are presented as mean ± SD, vitamin B12 and folic acid levels were compared with the normal laboratory range. CD: Celiac Disease, BMI: Body Mass Index, WBC: White blood Cell Count.

## Correlation between CCM parameters and clinical and laboratory measures

Age, duration of disease, height, BMI, tissue transglutaminase antibody level, haemoglobin, 25 OHD, vitamin B12, folic acid and iron levels did not correlate with CNFD, CNBD, CNFL, CNFT or IWL. Platelet count showed a significant inverse correlation with CNBD (B = -0.539; *P* = 0.017) and CNFL (B = -0.544; *P* = 0.016) (Fig 3A and 3B).

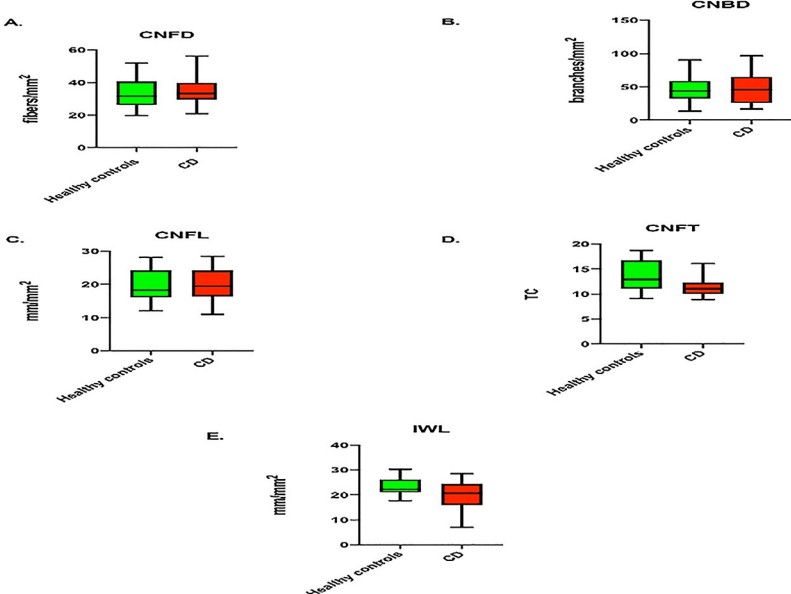

**Fig 2. CCM parameters in children with CD and healthy controls presented as box plots with mean and range.** (A) CNFD: Corneal nerve fiber density, (B) CNBD: corneal nerve branch density, (C) CNFL: corneal nerve fiber length, (D) CNFT: corneal nerve fiber tortuosity, (E) IWL: Inferior whorl length.

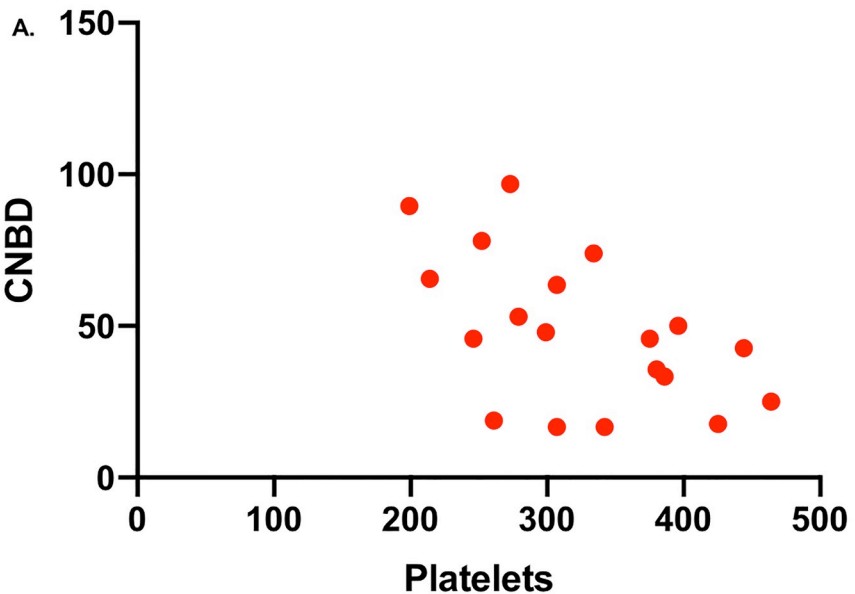

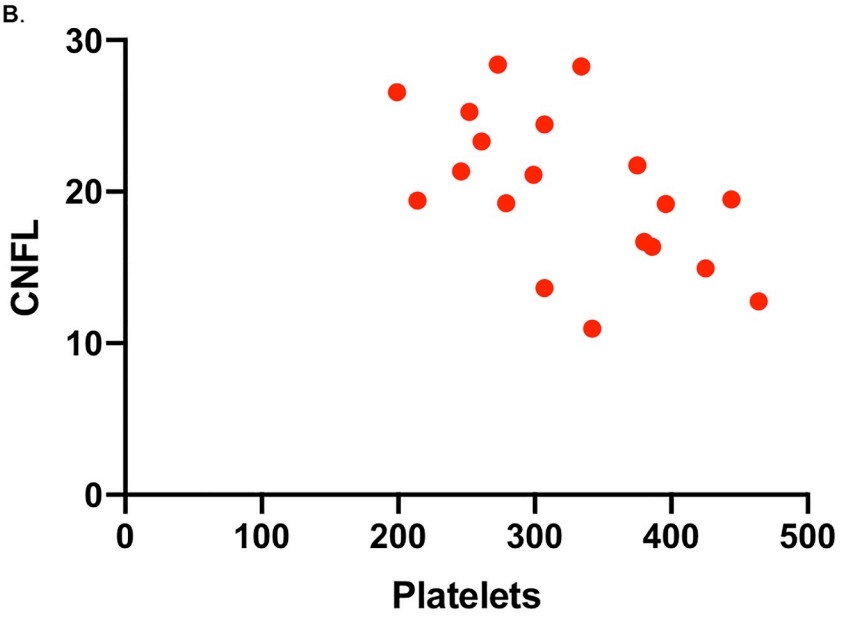

**Fig 3. Correlation plots between the platelet count with CNBD (A) and CNFL (B).** CNBD: corneal nerve branch density, CNFL: corneal nerve fiber length.

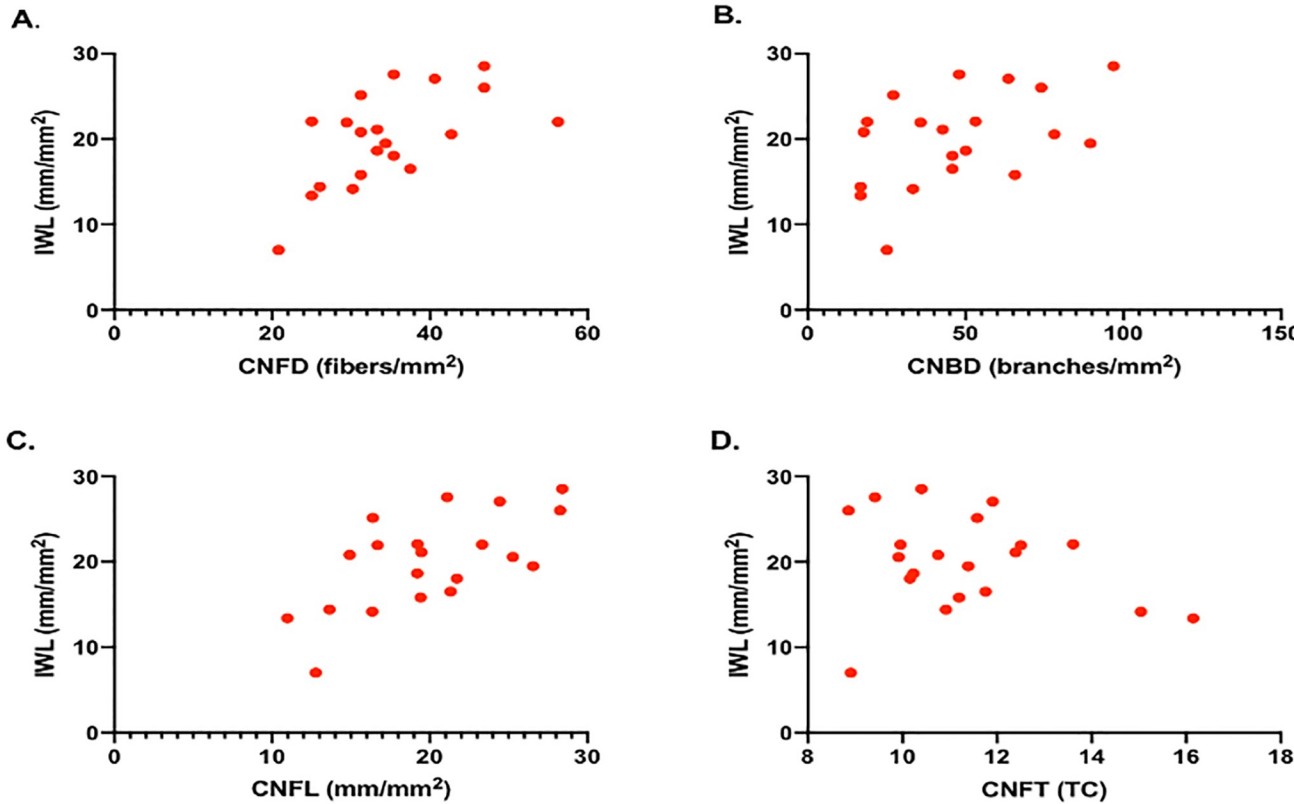

**Fig 4. Correlation plots between the IWL and CNFD (A), CNBD (B), CNFL (C) and CNFT (D).** IWL: Inferior whorl length, CNFD: Corneal nerve fiber density, CNBD: corneal nerve branch density, CNFL: corneal nerve fiber length, CNFT: corneal nerve fiber tortuosity.

### Correlation between central corneal and inferior whorl corneal nerve parameters

IWL correlated significantly with CNFD (B = 0.606; P = 0.005), CNBD (B = 0.459; P = 0.042) and CNFL (B = 0650; P = 0.002) but not with CNFT (B = -0.227, P = 0.337) (Fig 4A–4D).

## Discussion

This is the first study to assess corneal nerve morphology in children with celiac disease. We demonstrate no loss of corneal nerve fibres in the central cornea and a non-significant reduction in the inferior whorl length in children with CD. We believe it is critical to detect early sub-clinical nerve damage using sensitive techniques such as CCM to ensure compliance with a gluten-free-diet to prevent progressive nerve degeneration. However, our data indicate that children with CD do not have evidence of subclinical small nerve fibre damage. This is contrast to a nationwide population-based study of adults with biopsy verified-CD, where the prevalence of confirmed neuropathy was increased based on the International Classification of Disease (ICD) codes for any neuropathy, idiopathic neuropathy, Guillain-Barre Syndrome (GBS), chronic inflammatory demyelinating polyneuropathy (CIDP), mononeuritis multiplex and autonomic neuropathy [9]. Indeed, adults with CD have reported symptoms of burning, tingling, and numbness in the hands and feet which has been attributed to an underlying small fibre neuropathy [7]. Furthermore, abnormalities in electrophysiology and nerve biopsy [5,7],

reduced nerve conduction studies (NCS) [5,8] and abnormal QST [8] have been shown in adults with CD. Skin biopsy studies in adults with CD have also demonstrated periodic acid Schiff-positive and diastase resistance intracellular bodies in apocrine sweat glands [36], reduced intraepidermal nerve fiber density [37–39], increased axonal swelling and collateral reinnervation [40,41] in the calf and thigh [39]. Furthermore, sural nerve biopsy studies have shown a mild to moderately severe chronic axonopathy [7,42,43], focal inflammatory cell infiltrates in the epineurium and perivascular cuffing of the lymphocytes [42,43] with a patchy loss of myelinated fibers [42].

Peripheral neuropathy is associated with T1DM and CD; however, it is unclear whether T1DM patients with CD have an increased risk of neuropathy compared to either disease alone [44]. There is only one previous study which showed electromyographic evidence of axonal neuropathy in 7% of children with CD [10]. A case-report of a sural nerve biopsy from a child with CD demonstrated reduced unmyelinated fiber density, indicative of an early small fiber neuropathy [45]. The present study shows no reduction in central corneal nerve fibres in children with CD compared to healthy controls using CCM. This indicates that children with CD do not show a reduction in corneal nerves comparable to children with T1DM [11,31,46]. Although gluten neuropathy is common in adults with CD [4], in children with CD we show no evidence of corneal nerve loss. In a recent systematic review, age was the main driver of neuropathy in patients with CD [4]. Furthermore, we have observed a significant lower corneal nerve fibre tortuosity. This is in contrast with the observation that corneal nerve fibre tortuosity was increased in patients with diabetes [47,48]. This further supports a lack of evidence for small fibre degeneration in children with CD. However, the inferior whorl which is more distal to the central corneal nerves has shown an early loss in adults with diabetes [33,49–51], particularly in those with painful diabetic neuropathy [52,53]. We show a trend towards a reduction in IWL in children with CD compared to healthy controls and may be clinically relevant as it may help to identify children with very early nerve damage, particularly as 20% had a reduction which was >2SD of the mean in controls. The correlation between IWL and central corneal nerve measures is in agreement with a previous study in adults [49] and further supports a common underlying basis for corneal nerve loss at the centre and inferior whorl.

There were no associations between CCM measures and clinical variables such as BMI and vitamin D even though these were lower in CD and there was no association with tissue transglutaminase antibody levels or duration of disease. Whilst there was no difference in platelet density between children with CD and controls, paradoxically there was a significant inverse correlation with corneal nerve fibre measures. Previous studies have not shown a relationship between platelet density and corneal nerves or neuropathy. Platelets are rich in platelet derived growth factor which promotes the chemotaxis of fibroblasts, monocytes and macrophages and also inhibits metalloprotease activity and decreases inflammation [54]. Indeed, platelet lysate has been shown to improve corneal nerve fibre density and symptoms in patients with dry eye [55].

A limitation of the current study is the cross-sectional design and the small cohort size which may have resulted in the study being underpowered. Larger longitudinal studies are warranted, given the trend for a reduction in IWL suggesting very early distal nerve fibre loss in CD.

## Conclusion

We conclude that there is minimal evidence of small nerve fibre damage in children with CD.

## Supporting information

**S1 File. CCM and clinical data-PLOS ONE.**
(XLSX)

## Author Contributions

**Conceptualization:** Rayaz A. Malik, Anthony K. Akobeng.

**Data curation:** Hoda Gad, Saras Saraswathi, Bara Al-Jarrah, Ioannis N. Petropoulos, Georgios Ponirakis, Adnan Khan, Parul Singh, Souhaila Al Khodor, Mamoun Elawad, Wesam Almasri, Hatim Abdelrahman, Khalid Hussain, Mohamed A. Hendaus, Fatma Al-Mudahka, Khaled Abouhazima, Paraic McGrogan, Anthony K. Akobeng.

**Formal analysis:** Hoda Gad.

**Investigation:** Hoda Gad.

**Project administration:** Hoda Gad.

**Supervision:** Rayaz A. Malik, Anthony K. Akobeng.

**Writing – original draft:** Hoda Gad.

**Writing – review & editing:** Saras Saraswathi, Bara Al-Jarrah, Ioannis N. Petropoulos, Georgios Ponirakis, Adnan Khan, Parul Singh, Souhaila Al Khodor, Mamoun Elawad, Wesam Almasri, Hatim Abdelrahman, Khalid Hussain, Mohamed A. Hendaus, Fatma Al-Mudahka, Khaled Abouhazima, Paraic McGrogan, Rayaz A. Malik, Anthony K. Akobeng.

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
