## [Decision Letter · Decision Letter 0]

4 Aug 2020

PONE-D-20-16626

Corneal Confocal Microscopy Demonstrates Minimal Evidence of Distal Neuropathy in Children with Celiac Disease

PLOS ONE

Dear Dr. Malik,

Thank you for submitting your manuscript to PLOS ONE. After careful consideration, we feel that it has merit but does not fully meet PLOS ONE’s publication criteria as it currently stands. Therefore, we invite you to submit a revised version of the manuscript that addresses the points raised during the review process.

The reviewers see merit in this research, but clearer aims and more detailed methodology is required, especially to justify the sample size.

We look forward to receiving your revised manuscript.

Kind regards,

James Wolffsohn, PhD

Academic Editor

PLOS ONE

Journal Requirements:

2. Please provide additional details regarding participant consent. In the Methods section, please ensure that you have specified what type of consent you obtained (for instance, written or verbal) and whether the ethics committee approved this consent procedure. If verbal consent was obtained please state why it was not possible to obtain written consent and how verbal consent was recorded.

3.Thank you for stating the following in the Acknowledgments Section of your manuscript:

[This publication was made possible by a Sidra Internal Research Fund (SIRF) and a Biomedical Research Program [BMRP- 5726113101] grant from the Qatar National Research Fund (QNRF).]

 [The funders had no role in study design, data collection and analysis, decision to publish, or preparation of the manuscript.]

Reviewers' comments:

Reviewer's Responses to Questions

**Comments to the Author**

1. Is the manuscript technically sound, and do the data support the conclusions?

Reviewer #1: Partly

Reviewer #2: Yes

Reviewer #3: Yes

2. Has the statistical analysis been performed appropriately and rigorously? 

Reviewer #1: Yes

Reviewer #2: Yes

Reviewer #3: Yes

3. Have the authors made all data underlying the findings in their manuscript fully available?

Reviewer #1: Yes

Reviewer #2: Yes

Reviewer #3: Yes

4. Is the manuscript presented in an intelligible fashion and written in standard English?

Reviewer #1: Yes

Reviewer #2: Yes

Reviewer #3: Yes

5. Review Comments to the Author

Reviewer #1: Abstract

The authors state that they are establishing the prevalence of small fibre damage and neuropathy in children with celiac disease. Given their sample size, are they confidently able to establish PREVALENCE?

Introduction

The introduction is clear and well-written. I would again like to be convinced that what the authors are really doing is looking at prevalence. My understanding is that the authors are establishing CCM parameters in a group of children with celiac disease, compared to a group of controls. The presentation of the results does not support the aim of establishing prevalence. To do that, the authors would need to have an established cut-off for neuropathy and then establish neuropathy as the number of children with neuropathy as a proportion of the total number of children.

Methods

Please provide a sample size calculation. Lines 110-111 suggest that this is not possible, but I would argue that if you are detected neuropathy vs. no neuropathy, that there are many studies published that can provide means and standard deviations for this calculation.

Please explain how the six images were chosen e.g. were they overlapping by no more than 20% according to Vagenas et al?

Why were corneal nerve parameters quantified manually using CCMetrics, rather than ACCmetrics?

How was corneal tortuosity established?

The authors provide no information on the CCM technique, or on the other clinical measurements taken – I suggest that more detail be provided.

Results

Recommend the presentation of nerve data being to one decimal place – the additional decimal place does not add value. The same applies to p values, except where p < 0.01

The authors report correlations between corneal nerve features – I am not sure that this is required given the original study aims – I cannot see how this adds value.

Line 117 – please define OHD

Line 136-7: suggest including the scatter plot of this significant correlation

Discussion

I am concerned that the study was underpowered for the analysis undertaken

Figures

These are appropriate

Tables

These are appropriate

Reviewer #2: The study presented CCM may serve as potential surrogate biomarker in detecting early sign of distal neuropathy in children with CD, and indeed, this has never been reported in the literature. Overall, the objectives, plans and methods carried out were coherent, and analysis employed was statistically sound. However, there are certain parts of the manuscript that require amendments and further clarification as listed below:

- Line 54: Insert ‘and’ between “osteoporosis, arthritis”

- Line 81: Please define T1DM as it is the first time being introduced in the manuscript.

- Line 91-92: Participant with “previous corneal trauma or systemic disease affecting the cornea” was excluded from the study. What about ocular disease/disorders/surgery; e.g. keratoconus, pellucid marginal degeneration, LASIK surgery etc? If excluded please include that in the exclusion criteria.

- CCM procedures: The authors did not explain about the process took place in acquiring CCM images. Instrument, type of scanning, method of scanning etc. If you have set a standard protocol, please explain the procedures prior to image selection and quantification section. If the protocol has been published, please cite the paper appropriately.

- Image selection and quantification: What was the process involved in selecting those CCM images? Were there any criteria? What were they?

- Image selection and quantification: I assume you selected 6 images to analyse CNFD, CNBD, CNFL & CNFT, and additional 6 images to analyse inferior whorl length? If Yes, please rephrase the sentence as it is a bit confusing.

- Sample size: It is understandable that no prior data available for CCM in children with CD. However, could you please explain how you decide to examine 20 subjects in each group? What was the consideration?

- Line 117: Define 25 OHD as it is the first time being introduced in the manuscript.

- Line 121: Change (Fig 2A-D) to (Fig 2D), as this sentence only explains about CNFT, not other CCM parameters.

- Table 1: Add “mean±SD” below Healthy (n=20) and CD (n=20) to make it easier for readers to understand rather than writing it in the description (line 126).

- DISCUSSION: The first paragraph looks like a literature review rather than discussion. It is good to recap what has been previously reported, but please make sure it is not too lengthy. Perhaps, in the first paragraph, the authors can discuss about the results of the laboratory measures and relate with the current literature.

- The study discussed/compared the CCM data with previous reports on diabetic patients. It is known both conditions have an impact on peripheral nervous system. Hence, it would be good to have an opening statement in the second paragraph of the discussion about similar effects of both conditions (i.e. neuropathy), and that observable in the corneal nerves using CCM.

- However, please avoid stating a direct comparison as these two conditions are not exactly the same, for example: Line 168-169 “This is in contrast with our previous studies showing a significant reduction in central corneal nerves in children with T1DM”. I suggest use terms “similar trend” or “the results mimic”.

- Most CCM studies carried out on diabetic patients (cited in the manuscript) showed significant reduction in corneal nerve parameters. However, this was not the case for children with CD in the present study (e.g. CNFD, CNBD & CNFL). Why corneal nerve degeneration in CD was less apparent? Discussion about this is crucial to the paper.

- The study found CNFT was significantly lower in CD group compared to control. This is one of the hallmarks of the study but was not properly discussed (not even mentioned in the discussion). Please discuss what you think about it.

Reviewer #3: This paper shows a very good review of the topic generally and it is very well documented! It also presents a very throughout design for measurement and wide knowledge on the examination process. Only additional point would be to consider to have further follow up visits or even looking into separate age groups, like young adults, adults, etc... to confirm if any evolution? - I do however feel, this point is a bit reflected at the end of the discussion when stating "Larger longitudinal studies are warranted..." Good job!

6. PLOS authors have the option to publish the peer review history of their article (what does this mean?). If published, this will include your full peer review and any attached files.

Reviewer #1: No

Reviewer #2: No

Reviewer #3: **Yes: **Inma Perez-Gomez

---

## [Author Response · Author response to Decision Letter 0]

17 Aug 2020

August 13th, 2020

James Wolffsohn,

Academic Editor,

PLOS ONE

Re: PONE-D-20-16626 entitled "Corneal Confocal Microscopy Demonstrates Minimal Evidence of Distal Neuropathy in Children with Celiac Disease"

Dear Professor Wolffsohn,

Thank you for providing us with the opportunity to address the reviewers’ concerns.

Please find attached a point-by-point response (in bold) to the concerns raised and a revised manuscript with the changes highlighted in red. We hope that you find our responses satisfactory and that the manuscript is now acceptable for publication.

Kind Regards,

Rayaz A Malik, MD, PhD

Professor of Medicine,

Weill Cornell Medicine-Qatar,

Qatar Foundation,

Education City,

PO Box: 24144-Doha, Qatar

---

## [Editor Report · Decision Letter 1]

26 Aug 2020

Corneal Confocal Microscopy Demonstrates Minimal Evidence of Distal Neuropathy in Children with Celiac Disease

PONE-D-20-16626R1

Dear Dr. Malik,

We’re pleased to inform you that your manuscript has been judged scientifically suitable for publication and will be formally accepted for publication once it meets all outstanding technical requirements.

Kind regards,

James Wolffsohn, PhD

Academic Editor

PLOS ONE

Additional Editor Comments (optional):

Thank you for comprehensively addressing the reviewers comments.

---

## [Editor Report · Acceptance letter]

9 Sep 2020

PONE-D-20-16626R1

Corneal confocal microscopy demonstrates minimal evidence of distal neuropathy in children with celiac disease

Dear Dr. Malik:

I'm pleased to inform you that your manuscript has been deemed suitable for publication in PLOS ONE. Congratulations! Your manuscript is now with our production department.

Kind regards,

on behalf of

Professor James Wolffsohn 

Academic Editor

PLOS ONE